# Barriers and Facilitators to the Implementation of Evidence-Based Lifestyle Management in Polycystic Ovary Syndrome: A Narrative Review

**DOI:** 10.3390/medsci7070076

**Published:** 2019-06-27

**Authors:** Lucinda C. D. Blackshaw, Irene Chhour, Nigel K. Stepto, Siew S. Lim

**Affiliations:** 1Monash Centre for Health Research and Implementation, School of Public Health and Preventive Medicine, Monash University, Clayton, VIC 3168, Australia; 2Institute for Health and Sport, Victoria University, Melbourne, Victoria 8001, Australia; 3Australian Institute for Musculoskeletal Science (AIMSS), Victoria University, St. Albans, VIC 3021, Australia; 4Medicine-Western Health, Faculty of Medicine, Dentistry and Health Science, Melbourne University, Melbourne, VIC 3000, Australia

**Keywords:** weight management, exercise and physical activity, behaviour change, models of care, healthcare professionals and systems

## Abstract

Polycystic ovary syndrome (PCOS) is a complex condition that involves metabolic, psychological and reproductive complications. Insulin resistance underlies much of the pathophysiology and symptomatology of the condition and contributes to long term complications including cardiovascular disease and diabetes. Women with PCOS are at increased risk of obesity which further compounds metabolic, reproductive and psychological risks. Lifestyle interventions including diet, exercise and behavioural management have been shown to improve PCOS presentations across the reproductive, metabolic and psychological spectrum and are recommended as first line treatment for any presentation of PCOS in women with excess weight by the International Evidence-based Guideline for the Assessment and Management of Polycystic Ovary Syndrome 2018. However, there is a paucity of research on the implementation lifestyle management in women with PCOS by healthcare providers. Limited existing evidence indicates lifestyle management is not consistently provided and not meeting the needs of the patients. In this review, barriers and facilitators to the implementation of evidence-based lifestyle management in reference to PCOS are discussed in the context of a federally-funded health system. This review highlights the need for targeted research on the knowledge and practice of PCOS healthcare providers to best inform implementation strategies for the translation of the PCOS guidelines on lifestyle management in PCOS.

## 1. Introduction

Polycystic ovary syndrome (PCOS) is a complex condition involving reproductive, metabolic and psychosocial components. The diagnosis of PCOS is based on criteria encompassing the common presenting features of the condition; hyperandrogenism, oligo/ anovulation or menstrual disturbances and polycystic ovaries [1]. Insulin resistance (IR) underpins many of the physical and biochemical sequalae of PCOS [2,3,4,5]. Women with PCOS have higher body mass index (BMIs) than the general population while a reduction in weight has been shown to improve insulin resistance and PCOS symptoms [6,7,8,9]. The long term consequences of PCOS include cardiovascular disease (CVD), type II diabetes mellitus (DM2), sleep apnoea, psychological complications and low self-esteem as well as reproductive, fertility and pregnancy complications [10]. PCOS itself accounts for 19 to 28% of DM2 in reproductive aged women [11]. At the most recent estimate, PCOS was expected to cost the Australian economy 400 million dollars conservatively [10]. Majority of this cost stems from expensive fertility treatments and the long term chronic sequalae of IR and metabolic syndrome including DM2 and CVD [12]. Lifestyle management including diet and exercise with or without concurrent behavioural intervention has been shown to improve outcomes across the spectrum of the condition [9,13,14,15,16,17,18]. As a result, the International Evidence-based Guideline for the Assessment and Management of Polycystic Ovary Syndrome 2018 recommended healthy lifestyle behaviours for all women with PCOS regardless of presenting symptoms [19].

Given the range of symptoms of PCOS, a number of disciplines may be involved in PCOS patient care including general practice, allied health, and medical specialties. Lifestyle management has the potential to improve symptoms managed by each of these disciplines, however evidence regarding its implementation is lacking. In order to best provide care, barriers and facilitators to adequate and comprehensive provision of evidence-based lifestyle management for PCOS by health professionals must be understood.

## 2. Pathogenesis and Aetiology

The pathophysiology of PCOS is complex (Figure 1) with its aetiology remaining unclear. IR and compensatory hyperinsulinemia are known to be key components of the syndrome, affecting 75% of lean PCOS affected individuals and 95% of obese individuals [2,3]. The degree of insulin resistance experienced by women with PCOS has also been shown to be above what would be expected given their BMI [2,5]. Insulin resistance affects both hyperandrogenaemia and ovulation by acting on the pituitary and ovaries arresting follicular development and increasing androgen production [20,21]. Insulin further inhibits the production of hepatic sex hormone binding globulin (SHBG) at the liver contributing to the higher serum concentration of free androgens commonly seen in PCOS [22]. Decreasing IR through weight loss, lifestyle interventions or with insulin sensitising agents improves the clinical features of PCOS while excess weight worsens IR and symptom severity [3,9,23,24].

## 3. Obesity and Polycystic Ovary Syndrome

It is well recognised that women with PCOS are more likely to be overweight or obese which may reflect the impact weight has on symptom severity [25,26]. Obesity amongst women with PCOS has risen with population increases in prevalence, from 51% to 74% in the years between 1987 and 2002 [27]. Obesity exacerbates many of the symptoms of PCOS including IR, hyperandrogenism and menstrual disturbances [3,23,28]. Women with PCOS are more likely to be overweight, obese and centrally obese compared to women without PCOS [29,30]. Teede et al. found women with PCOS gain on average 2.6 kg more than their non-PCOS counterparts over a 10 year period [29]. Weight gain and obesity contribute to negative emotional wellbeing and continues to be a key concern for women with the condition [31,32].

A number of mechanisms may contribute to the higher rates of obesity and difficulty losing weight in women with PCOS. Biological mechanisms such as altered metabolic flexibility, altered skeletal muscle signalling and fat storage, brown adipose tissue thermogenesis and increased appetite have been implicated while gut hormones responsible for regulating appetite such as cholecystokinin (CKK) and ghrelin may also be dysregulated [4,33,34,35,36,37,38,39,40]. Behavioural differences between women with and without PCOS may also explain the increased rate of obesity and weight reduction difficulties. Survey evidence indicate these women’s daily diets tend to contain an additional 250 kJ compared to women without PCOS [33]. Sedentary behaviour is also documented to be greater in women with PCOS than the general public [33].

## 4. Management of Polycystic Ovary Syndrome

Management of PCOS utilises a diverse range of strategies due to the intersection of many specialties managing the syndrome [41,42,43]. However, the International Evidence-based Guideline for the Assessment and Management of Polycystic Ovary Syndrome 2018 endorse lifestyle modifications as first-line treatment for all symptom presentations [19]. Lifestyle management programmes involve a combination of dietary modifications, increased physical activity and behavioural adjustments to achieve weight loss and/or maintenance [44].

## 5. Lifestyle Management

A summary of the various aspects of lifestyle management interventions and their benefits on anthropometric, cardiovascular, metabolic, reproductive and psychological health, based on evidence from meta-analyses, is shown in Table 1. The meta-analyses were identified using search terms “lifestyle” or “diet” or “exercise” and “PCOS” using PubMed, Scopus and the Cochrane database of systematic reviews, through expert referral and previous reviews with key publications selected for their relevance by first authors.

## 6. Weight Management

Weight management is prioritised by health professionals and patients with PCOS. Qualitative evidence published by Love et al. reported weight management was sought by PCOS women who had concerns about their general health, fertility and appearance [45]. For overweight women with PCOS, lowering body mass by 5% can reduce serum insulin and testosterone levels, and improve menstrual and reproductive function [46,47,48]. Weight loss also improves depression and PCOS-specific quality of life scores [49]. Thus, effective weight loss strategies are the focus of research. Lifestyle interventions are capable of managing these immediate health concerns, as well as preventing long term chronic diseases. Lifestyle interventions for prevention of chronic disease in other high risk populations have been shown to be cost-effective, as associated costs are offset by reductions in non-intervention related medical care [50].

Long term weight management does present challenges. In the general population, weight is poorly maintained with few sustaining long-term weight loss [51,52]. Complex social, contextual, cognitive and emotional obstacles challenge the maintenance of weight loss [53]. Women with PCOS have greater drop-out rates in lifestyle intervention studies [54,55]. However high attrition in lifestyle intervention studies are not uncommon for females and younger participants [56]. Psychological factors contributing to poor trial adherence such as disordered eating, anxiety and depression could explain difficulties these women experience maintaining weight [57]. Life stage-specific factors and priorities have been barriers for younger participants adhering to lifestyle interventions [58,59]. For women with PCOS who have excess weight, there are additional barriers to overcome that are not observed in women without PCOS. These include a perception of lower self-worth which is related to the impact of the condition on physical appearance [45]. Hyperandrogenemia and menstrual abnormalities; common features in PCOS were found by Lim et al. to be associated with increased cravings for high fat and fast foods. Psychological distress, another common comorbidity in women with PCOS, was associated with cravings for sweet and high fat foods [60]. These cravings may reduce the likelihood of long-term weight maintenance.

## 7. Diet Intervention

The International Evidence-based Guideline for the Assessment and Management of Polycystic Ovary Syndrome 2018 recommend that a variety of dietary approaches should be utilised to reduce dietary energy intake, while incorporating general healthy eating principles [19]. Dietary management has been used alone and in combination with exercise and behavioural therapy for weight loss and management in PCOS [8,48,61,62]. To achieve weight loss through diet modification in women with PCOS, there should be a negative energy balance of approximately 30% to achieve an energy deficit of 500–750 kcal per day [19]. These recommendations are consistent with general population guidelines that suggest low-calorie diets containing sufficient carbohydrates and proteins with low-fat intake will provide optimum results [63]. Diet advice provided to women with PCOS often specifies a reduction in dietary glycaemic index in addition to overall calorie reduction to induce weight loss [64].

All diets that aim to reduce weight are considered beneficial [19]. The absence of prescribed dietary interventions serves as an opportunity for healthcare providers to tailor dietary modifications towards the patient’s individual needs and preferences as no diet is known to be conclusively more effective in PCOS [65].

## 8. Physical Activity

Physical activity (any bodily movement produced by skeletal muscles that requires energy expenditure) and exercise (activity requiring physical effort, carried out to sustain or improve health and fitness) for all women with PCOS has been shown to improve presentations of PCOS both through weight loss and independently of weight loss [66,67,68]. To prevent weight gain and maintain health, International Evidence-based Guideline for the Assessment and Management of Polycystic Ovary Syndrome 2018 recommend that different durations and intensities of exercise are to be met for different age groups [19,69]. These recommendations are in accordance with those for the general population. Adults aged 18–64 should perform physical activity for a minimum of 150 min per week of moderate intensity, 75 min per week of vigorous intensity or an equivalent of both [19,69]. Recommendations for adults include muscle-strengthening activity on two non-consecutive days [19,69]. For adolescents, the recommendations for physical activity are more intensive, recommending a minimum of 60 min per day most days of the week [19,69]. The physical activity should be moderate to vigorous intensity and complemented with muscle-strengthening activities performed at least three times per week [19,69]. In general, minimised sedentary, screen and sitting time are advised [19,69]. Regular exercise or physical activity is considered important for women with PCOS, but evidence from the general population suggests exercise conjunction with a hypocaloric diet has a greater ability to increase weight loss [70].

## 9. Behavioural Intervention

The International Evidence-based Guideline for the Assessment and Management of Polycystic Ovary Syndrome 2018 recommend the inclusion of behavioural strategies in lifestyle interventions to optimise weight management, improve general health and maintain emotional wellbeing in PCOS women [19]. Behavioural therapy aims to modify lifestyle behaviours and involves implementing goal setting, self-monitoring, identifying barriers and problem solving [57,71]. These strategies are used in conjunction with diet modification and exercise therapy to increase program adherence and efficacy [72]. These are often provided by a health professional such as a psychologist [71]. When ongoing contact with health professionals is combined with social support, behavioural therapy is often associated with stronger positive outcomes in the general population [73]. Adolescents with PCOS provided with nutrition counselling and behavioural therapy experienced greater weight loss compared to those receiving neither however the precise nature of the intervention was not reported [61]. Given their efficacy in the general population they are also likely to result in beneficial outcomes for women with PCOS.

## 10. The Role of Healthcare Professionals in Lifestyle Management in Polycystic Ovary Syndrome

A number of different medical and allied health practitioners are likely to be involved in PCOS care across different life stages, including general practitioners, allied health professionals and specialists such as obstetrician and gynaecologists and endocrinologists [74]. Weight and lifestyle management are beneficial across all health professional symptom domains thus each role represents an opportunity to engage in lifestyle management.

However, it should be noted women with PCOS have reported dissatisfaction with provision of care and support from physicians when compared to women without PCOS [32,75]. A 2018 study concluded that women with PCOS report overall greater distrust in the opinions of their primary care physicians and reportedly had more arguments with other healthcare professionals, compared to women without PCOS [75]. This may impact the ability of PCOS care providers to effectively implement lifestyle management.

## 11. General Practitioners

General practitioners (GPs) are responsible for the initial diagnosis, treatment of less complex issues and overseeing continuity of care in chronic diseases [76]. In PCOS, provision of evidence-based resources, lifestyle intervention and psychological health should be effectively managed in primary care [77].

The role of the GP as the coordinator of patient care means that they are more likely to have regular contact and ongoing relationships with patients compared with other medical specialties and allied health professionals. This places GPs in an ideal position to facilitate long term lifestyle and weight management. The role of the GP in primary care allows discretionary prescription of lifestyle management themselves, or referral to the appropriate medical or allied health specialist if it is beyond their expertise. However, evidence suggests GPs do not regularly engage in lifestyle management. One Australian survey identified less than one third of overweight and obese patients reported their GP mentioning lifestyle management in the preceding twelve months [78]. Patients also do not appear to have confidence that GPs will be able to manage their weight effectively [79]. However, these were derived from the general population and we do not have information specific to the lifestyle management of PCOS.

## 12. Allied Health

Allied health professionals are essential in the collaborative management of complex and chronic conditions such as PCOS [76]. Each allied health professional can provide their own expertise on management of PCOS as part of multidisciplinary care that is prioritised by the International Evidence-based Guideline for the Assessment and Management of Polycystic Ovary Syndrome 2018 [19]. Lifestyle management can be outsourced by other medical professionals to allied health professionals to support the intensive and complex lifestyle and weight management interventions [19]. Allied health professionals involved in PCOS care often include, but are not limited to, dieticians, exercise physiologists, physiotherapists and psychologists [19,69].

When allied health are utilised in a multidisciplinary PCOS setting, participants appear to have better weight loss results. Geier et al. found that nearly 70% of adolescents with PCOS demonstrated weight loss or maintenance when supported by a health psychologist and dietitian as part of a multidisciplinary team however as discussed earlier this intervention was not well described [61]. Dieticians and exercise physiologists can educate and recommend a patient-specific diet and exercise plan tailored for weight loss or prevention of weight gain [64,80]. Allied health services appear underutilised in the management of PCOS in many countries [64]. A 2005 survey of UK dietitians and women with PCOS by Jeanes et al. reported that although most women were following a diet, less than 15% had consulted a dietitian, and a major component of diet advice was collected from less reputable sources including books and the internet [64].

Psychologists could also be involved in the management of the psychological and psychosocial concerns common in women with PCOS [19]. Trained psychologists could also deliver behavioural support in lifestyle modification, although the extent to which psychologists engage with lifestyle management in PCOS is not known.

## 13. Obstetricians-Gynaecologists

The role of obstetricians-gynaecologists (OBGYNs) in PCOS predominantly concerns the management of menstrual irregularities and infertility. Although a majority of OBGYNs consider lifestyle advice as first line management for non-fertility reasons in women with PCOS, the proportion considering this management as first line is lower than the proportion that do in other specialities [42,43,81]. Greater provision of healthy lifestyle advice is needed as the International Evidence-based Guideline for Assessment and Management of Polycystic Ovary Syndrome 2018 recommends that healthy lifestyle advice to be recommended to all patients with PCOS, regardless of other presentations [19].

## 14. Endocrinologists

Endocrinologists form an integral part of the PCOS care team over a lifetime given that PCOS is an endocrinopathy (Figure 1) with inherent risks of obesity, metabolic syndrome, CVD, IR, DM2 and dyslipidaemia. Past studies have consistently shown endocrinologists are more likely to discuss lifestyle management for both fertility and non-fertility reasons when compared to their OBGYN counterparts [42,43,81]. However, lifestyle advice is often deemed to be unsatisfactory or general and inadequate by patients [82,83]. Advice may only be provided to women who appear overweight meaning health professionals are missing opportunities to prevent weight gain [82]. This is of particular importance given women with PCOS gain more than their reproductive aged peers, a population group already predisposed to significant longitudinal gains [29,84].

## 15. Barriers and Facilitators for Evidence Based Lifestyle Management

Despite evidence supporting lifestyle management in PCOS and endorsement by the International Evidence-based Guideline for the Assessment and Management of Polycystic Ovary Syndrome 2018 there exist a number of barriers to the implementation of evidence-based lifestyle management at both the systemic and individual practitioner level. The barriers discussed below are mostly within the context of federally-funded healthcare systems as exists in Australia. The studies included in this review have been conducted in a number of developed countries including the USA, Australia, Ireland, the Netherlands, Portugal and Singapore, where referral pathways and funding structures likely differ. Despite the differences in healthcare systems, there appears to be common barriers to lifestyle management across these countries, highlighting clinical knowledge gaps on the implementation of lifestyle management within healthcare systems in developed countries. It is even more concerning when no published data exists for developing nations grappling with the similar issues for women with PCOS.

In general, many PCOS patients report disappointment with information provided to them about lifestyle management [32,83,85]. Furthermore, the “silo-ed” nature of a medical system fragmented by specialty has been blamed for some of the discontent among women with PCOS [74,86]. Lack of communication between health professionals results in delays to diagnosis and treatment, insufficient use of allied health and inadequate information provision exacerbating dissatisfaction in this patient population [32,83,85].

The barriers to the implementation of effective lifestyle management are not known indicating the need for targeted research among PCOS healthcare providers. Much of the literature focuses on barriers and facilitators of lifestyle intervention experienced by GPs which is consistent with the perception that lifestyle management is the responsibility of the GP [77]. In the wider population systemic barriers such as time pressure and insufficient financial reimbursement are commonly cited barriers to the provision of lifestyle and obesity management in primary care [87,88,89,90,91,92,93,94]]. Lack of clarity in healthcare roles appears to be an issue for coordination of lifestyle management between other healthcare providers [93]. In PCOS others have also implicated long waiting times for allied health providers as catalysts for patient’s accessing alternative sources of health information including potentially unreliable information on the internet and in books [32]. Clinicians’ perception of patient access to and the availabilty of allied health professionals has impacted on their decision to discuss lifestyle management in the general population [88,95,96]. GPs were unlikely to refer patients to allied health professionals to support lifestyle management if they perceived that the consultations were expensive, the provider was located too far away and/or inaccessible by public transport [95,96].

Ko et al. when assessing barriers to lifestyle management in a multidisciplinary PCOS setting identified other systemic barriers to lifestyle management implementation in PCOS. These included short consultations, non-standardised delivery of lifestyle management, lack of enforcement and follow up of interventions and unsatisfactory coordination with allied health staff [97]. Further, the absence of culturally appropriate resources meant when advice was provided it was not as effective for some patient groups [97]. This study was part of a quality improvement framework for a PCOS fertility clinic in Singapore and the barriers reported may not be easily translated to other settings.

General pratcitioners are often held in high esteem by patients which provides them great influence and opportunity to reinforce health messages [92]. Yet, past surveys of healthcare professionals have described how health professionals, find it difficult to provide guidance on lifestyle management due to perceived lack of expertise and training [87,88,91,92,93]. GPs are less inclined to engage in lifestyle management if they percieve a lack of motivation and responsibility for health in their patients [87,89,90,91,92,93,95,96,98,99]. Some healthcare professionals report a lack of motivation to implement lifestyle management as poor patient compliance means repeated messages made during lifestyle consultations are unlikely to prompt changes [88,89,93,98]. In both PCOS and the general population, practitioners have reported patients themselves lack the time and priority for lifestyle interventions [97,98]. These patients were more likely to desire or be offered pharmaceutical treatments [97,98].

Fear of deterioration of patient relationship and offending patients makes it difficult for clinicians to initiate lifestyle consultations as discussing sensitive topics like weight management can be considered judgemental [92,93,99]. The provider-patient partnership can also be strained by clinicians’ own experiences and expectations regarding lifestyle management [93,95]. A study by Kim et al. reported high expectiations may often result in clinicians pushing patients too hard for desired results and deteriorating the vital patient-provider partnership [95]. Although patients’ own unrealistic expectations make lifestyle management difficult to implement [89]. Women with PCOS have detailed a perceived lack of empathy from doctors in discussions of weight and some clinicians admitted they found it difficult to empathise thus supporting lifestyle management is difficult [32,93].

A summary of common barriers cited in studies assessing clinician attitudes to obesity and weight management are provided in Table 2. These are not PCOS-specific and are unlikely to reflect the unique issues experienced by clinicians managing women with the condition. Further barriers specific to the implementation of lifestyle management in a PCOS context need to be better understood to best address these issues.

## 16. Knowledge and Practice Gaps

This review has identified a number of knowledge gaps within the available literature implicating a strong need for further research. Only one study reported PCOS-specific barriers, while only two reported on barriers experienced by specialists. Thus, whether the barriers discussed in this review apply outside of general population primary care remains purely speculative. Barriers and facilitators have been discussed in the context of a federally-funded healthcare system such as exists in Australia and thus the recommendations from this review will not be uniformly relevant for all international contexts. Future research should aim to identify which women with PCOS are receiving lifestyle advice and if this advice conforms to recommendations made by the International Evidence-based Guideline for the Assessment and Management of Polycystic Ovary Syndrome 2018. Furthermore, there needs be greater insight into the type of advice and support women with PCOS would appreciate to facilitate their adherence to lifestyle interventions. Knowing how to best support health professionals in providing this advice is also vital, thus barriers and facilitators to the implementation of lifestyle management for PCOS need to be identified.

## 17. Conclusions

While the benefits of lifestyle interventions in women with PCOS have been established and endorsed by the International Evidence-based Guideline for the Assessment and Management of PCOS 2018, this review highlights the potential barriers to its the implementation and translation into clinical practice. The extent to which providers are prescribing lifestyle management for PCOS is not well described. Whether the provision of lifestyle management is according to evidence is not known. Furthermore, barriers to the implementation of these lifestyle recommendations for women with PCOS that are faced by various health professionals has not been explored outside of one Singaporean study. Further research should seek to address these unknowns to facilitate translation of guideline recommendations for lifestyle management into clinical practice and thus better outcomes for women.

## Figures and Tables

**Figure 1 medsci-07-00076-f001:**
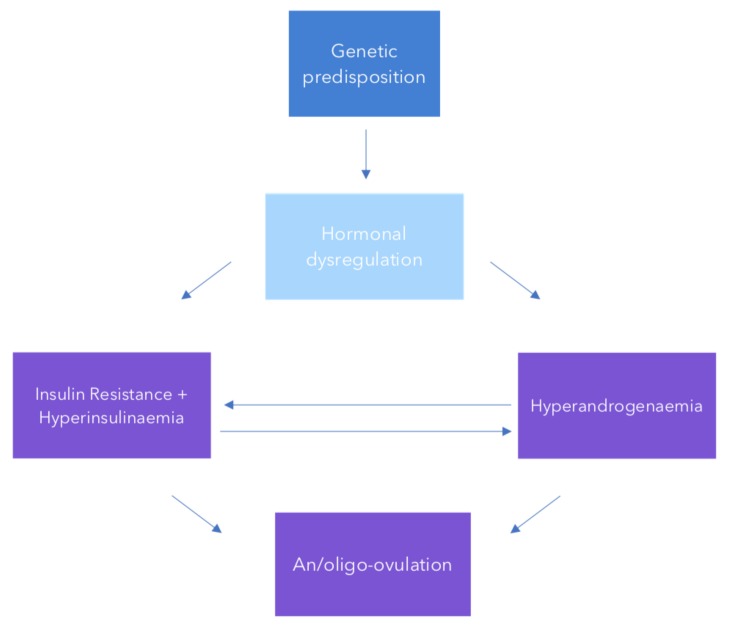
A schema of the pathogenesis and aetiology of polycystic ovary syndrome (PCOS).

**Table 1 medsci-07-00076-t001:** A summary of current evidence from meta-analyses on lifestyle management in PCOS.

Author	Intervention Type	Control	Number of Included Papers	Total Number of Participants (*n*)	Population and PCOS Diagnostic Criteria	Key Findings
Moran et al. [9]	Structured dietary, exercise or behavioural intervention or any combination	Minimal intervention	6	*n* = 164	PCOS (Rotterdam or NIH)	Anthropometric: ↓ weight, waist, WHR; NC BMI
						Cardiovascular-Respiratory: NA
						Metabolic: ↓OGTT insulin, FI; NC glucose, cholesterol
						Reproductive: ↓ TT, hirsutism; NC FAI, SHBG
						Psychological: NA
Domecq et al. [13]	Diet, physical exercise or combination diet and physical exercise	Metformin or minimal intervention	10	*n* = 610	PCOS (criteria unspecified)	*Lifestyle vs. Minimal intervention*
						Anthropometric: NA
						Cardiovascular-Respiratory: NA
						Metabolic: ↓ FBG, FBI, direct correlation between BMI and FBG, no significant trend between BMI and FBG
						Reproductive: ↓ FGS
						Psychological: NA
						*Lifestyle vs. Metformin*
						Anthropometric: NA
						Cardiovascular-Respiratory: NA
						Metabolic: NSD FBG, FBI
						Reproductive: NSD hirsutism score pregnancy rate
						Psychological: NA
Haqq et al. [14]	Physical exercise alone or combination diet and physical exercise	Usual care i.e., No active intervention, metformin, untreated controls, placebo, healthy diet only	7	*n* = 206	PCOS (criteria unspecified)	*Lifestyle Intervention (Combinatorial)*
						Anthropometric: NA
						Cardiovascular-Respiratory: NA
						Metabolic: NA
						Reproductive:↑ FSH, SHBG; ↓ TT, androstenedione, FAI, FGS; NSD LH
						Psychological: NA
						*Exercise Alone*
						Anthropometric: NA
						Cardiovascular-Respiratory: NA
						Metabolic: NA
						Reproductive:↑ FSH, SHBG, free testosterone; ↓ TT, androstenedione, FGS; NSD LH, FAI, E2, LH:FSH
						Psychological: NA
Haqq et al. [15]	Physical exercise alone or combination diet and physical exercise	Usual care Sedentary control, placebo, diet only, or metformin	12	*n* = 668	PCOS (criteria unspecified)	*Lifestyle intervention (Combinatorial)*
						Anthropometric: ↓ BMI, BM, WC, WHR, BF%
						Cardiovascular-Respiratory: ↑ VO2 max
						Metabolic: ↓ CRP; NSD insulin, glucose, IR, TG, TC, LDL, HDL
						Reproductive: NA
						Psychological: NA
						*Exercise alone*
						Anthropometric: ↓ BMI, WC
						Cardiovascular-Respiratory: ↓ RHR; ↑ VO2 max
						Metabolic: NA
						Reproductive: NA
						Psychological: NA
Naderpoor et al. [16]	Lifestyle and metformin or metformin alone	Lifestyle or lifestyle and placebo	12	*n* = 608	PCOS (Rotterdam)	Lifestyle + Metformin vs. Lifestyle ± Placebo 6/12
						Anthropometric: ↓ BMI, SAT; NSD WC, WHR, VAT
						Cardiovascular-Respiratory: NSD SBP, DBP
						Metabolic: NSD TC, HDL, LDL, TG, markers of IR, FBG
						Reproductive: ↑ menstrual frequency over 6/12 and 12/12; NC acne, FG score, FAI, SHBG, DHEAS, LH
						Psychological: NSD QOL
						*Lifestyle* *±* *Placebo vs. Metformin*
						Anthropometric: NSD BMI; ↓WC
						Cardiovascular-Respiratory: NA
						Metabolic: NSD FBG, insulin AUC, glucose AUC
						Reproductive: ↑ SHBG, TT; NSD FAI, menstrual cycles 6 to 12/12
						Psychological: NA
Benham et al. [17]	Aerobic exercise, resistance training, combination of aerobic and resistance training both with and without behavioural and diet interventions	Diet intervention alone or standard care	14	*n* = 617	PCOS (Rotterdam NIH, NIH Phenotypes, physician diagnosed and unreported)	*Meta-analysis*
						Anthropometric: ↓WC; NC BMI, BF%
						Cardiovascular-Respiratory: ↓ SBP; NC VO2 max, DBP
						Metabolic: ↓ FI, TC, LDL, TGs; ↑ HDL; NC FBG
						*Semi-quantitative Analysis*
						Reproductive: NC/↑ pregnancy ↑ovulation rate/cycles, menstrual frequency/regularity menstrual cycle length
						Psychological: NA
Kite et al. [18]	Exercise or	Control or diet alone	18	*n* = 758	PCOS (Rotterdam NIH, physician diagnosed)	*Exercise* vs. *Control—Change from Baseline*
						Anthropometric: NSD BMI, WHR, ↓WC, FM, FFM, BF%
						Cardiovascular-Respiratory: NSD SBP, DBP, VO_2_ max, RHR
	Exercise and diet	Control or diet alone				Metabolic: NSD FBG; ↓ FI, HOMA-IR, TC, LDL, TG; NC HDL
						Reproductive: NSD TT, SHBG, FT, FAI, FGS, E_2,_ DHEA-S, LH, FSH, LH:FSH ratio, PG, Prolactin, AMH, Adiponectin.
						Psychological: NA

AMH: anti-mullerian hormone; BF%: body fat %; BMI: body mass index; Corr(X,Y): direct correlation between X and Y; CRP: C-reactive protein; DBP: diastolic blood pressure; DHEAS: dehydroepiandrosterone sulfate; E_2_: oestradiol; FAI: free androgen index; FBG: fasting blood glucose; FI: fasting blood insulin; FFM: fat free mass; FGS: ferriman-gallwey score; FM: fat mass; FSH: follicle-stimulating hormone; FT: free testosterone; HDL: high density lipoprotein; HOMA-IR: homeostatic model of assessment-IR; IR: insulin resistance; LDL: low density lipoprotein; LH: luteinising hormone; NC: no change; NSD: no significant difference; OGTT: oral glucose tolerance test; PG: progesterone; QOL: quality of life; RHR: resting heart rate; SAT: subcutaneous adipose tissue; SBP: systolic blood pressure; SHBG: sex hormone binding globulin; TC: total cholesterol TG: triglycerides; TT: total testosterone; VAT visceral adipose tissue; VO_2_ max: maximal oxygen consumption; WC: waist circumference; WHR: waist-to-hip ratio; NIH: National Institute of Health; NA: not assessed.

**Table 2 medsci-07-00076-t002:** A summary of key barriers to implementation of lifestyle interventions experienced by healthcare professionals at the level of the health system and the individual.

Barriers to Lifestyle and Obesity Interventions	Studies with Identified Barrier(s)	Potential Facilitators and Solutions
Health System Level		
Lack of time	Kushner et al. [88] Campbell et al. [89] Nicholas et al. [87] Jansink et al. [93] Lambe et al. [92] Passey et al. [94] Salinas et al. [91] Epling et al. [90]	Greater team effort between clinicians, patients and policy makers to propose efficient and necessary time of consultations [88] Modifications made to fee-for-service systems to increase the reward for lifestyle counselling [87]
Lack of reimbursement	Kushner et al. [88] Passey et al. [94] Lambe et al. [92] Salinas et al. [91] Epling et al. [90] Ko et al. [97]	Ensure that amount of reimbursement available for lifestyle management reflects government health policy expectations [97]
Limited access to and availability of allied health providers and other members of the multidisciplinary care team	Nicholas et al. [87] Passey et al. [94] Epling et al. [90] Teixeira et al. [98] Glauser et al. [99] Kim et al. [95]	Recognition that multi-disciplinary facilities and schemes should be available for obese people with and without existing chronic diseases [79] Greater communication along clinician referral pathways as it is likely that there are higher referral rates (to co-located providers) when trust between providers is established [95]
Provider location	Kushner et al. [88] Kim et al. [95]	Increase service availability as health providers report being more likely to refer where services are readily available [95]
Lack of coordination between healthcare providers	Jansink et al. [93] Kim et al. [95]	Greater communication between providers to ensure individual roles are clear and management is effective [93]
Lack of high quality and affordable material for patient education	Jansink et al. [93] Kushner et al. [88]	More research into the efficacy of pamphlets and newsletters [88]
Health Professional Level		
Self-perceived lack of expertise and training	Kushner et al. [88] Nicholas et al. [87] Ampt et al. [96] Jansink et al. [93] Lambe et al. [92] Salinas et al. [91] Glauser et al. [99]	Updated training and continued professional development with increased focus on implementing concepts of lifestyle interventions into practice [88]
Perception of low patient motivation, responsibility, and/or compliance	Kushner et al. [88] Campbell et al. [90] Nicholas et al. [87] Ampt et al. [96] Lambe et al. [92] Jansink et al. [93] Salinas et al. [91] Epling et al. [90] Glauser et al. [99] Kim et al. [95] Teixeira et al. [98] Ko et al. [97]	Patients are more likely to adhere to advice if they can recall it. Pertinent information needs to be delivered clearly and concisely [88] Lifestyle interventions need to be tailored to meet patient’s skill capacity, accessibility to facilities and with support for possible psychosocial barriers [93] Implementation of patient centred motivational interviewing [93]
Poor patient-provider partnership	Jansink et al. [93]	Combatting false or high expectations of health providers to prevent resistance from patient [93]
Reluctance to or fear of offending patient	Jansink et al. [93] Lambe et al. [92] Glauser et al. [99]	Training in effective communication, particularly when involving sensitive issues such as weight [79]
Motivation to implement based on provider’s own interests, expectations and experiences	Jansink et al. [93] Kim et al. [95] Teixeira et al. [98]	Improved education and feedback to clinicians about interventions to negate personal opinions [95]
Limited patient time and priority to regularly attend consultations	Teixeira et al. [98] Ko et al. [97]	Utilisation of information technology-enabled lifestyle management tools [97] Increase opportunities for incidental activity [97]

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
