# Peer review of "Barriers and Facilitators to the Implementation of Evidence-Based Lifestyle Management in Polycystic Ovary Syndrome: A Narrative Review"

_medsci, 2019, doi:10.3390/medsci7070076_

Reviewer 1 Report

Overall this is a comprehensive narrative review detailing the challenges of implementing a lifestyle program in PCOS. As the authors note the recommendations are heavily based on practice in Australia and as such may not be uniformly applicable to other healthcare systems. This should be better emphasized in the discussion. Additional points:

How were the studies selected for inclusion in Table 1--how many were reviewed and excluded as this is not comprehensive?

Regarding the role of healthcare professionals, you should note the following reference:

A W Lin, E J Bergomi, J S Dollahite, J Sobal, K M Hoeger, M E Lujan, Trust in Physicians and Medical Experience Beliefs Differ Between Women With and Without Polycystic Ovary Syndrome, Journal of the Endocrine Society, Volume 2, Issue 9, September 2018, Pages 1001–1009, https://doi.org/10.1210/js.2018-00181

This concludes women with PCOS reported greater overall distrust in the PCP’s opinions and more arguments with their health care providers than women without PCOS, which may influence some of your recommendations.

Author Response

Dear reviewer 1 and editor,

Thank you for the constructive feedback. Please see below our response to your comments and suggestions:

Comments and suggestions

Responses

Reviewer 1

Overall   this is a comprehensive narrative review detailing the challenges of   implementing a lifestyle program in PCOS. As the authors note the   recommendations are heavily based on practice in Australia and as such may   not be uniformly applicable to other healthcare systems. This should be   better emphasized in the discussion.

Emphasis has been   added in the Discussion to highlight this limitation [page 10 line 330 - 333]:

“Barriers and   facilitators have been discussed in the context of a federally-funded   healthcare system such as exists in Australia and thus the recommendations   from this review will not be relevant for all international contexts.”

This focus on the   Australian health system has also been emphasised in the abstract [page 1 in   lines 27-29] “In this review, barriers and facilitators to the implementation   of evidence-based lifestyle management in reference to PCOS.” Replaced by “In   this review, barriers and facilitators to the implementation of   evidence-based lifestyle management in reference to PCOS are discussed in the   context of a federally-funded health system.”

And in the discussion  [page 6 line 262] with “The barriers   discussed below are mostly within the context of federally-funded healthcare   systems.  replaced by “The barriers   discussed below are mostly within the context of federally-funded healthcare   systems as exists in Australia.  “

How   were the studies selected for inclusion in Table 1-- How many were reviewed   and excluded as this is not comprehensive?

For clarity the text   has been updated to below [page 5 lines 100-102]:

A   summary of the various aspects of lifestyle management interventions and   their benefits on anthropometric, cardiovascular, metabolic, reproductive and   psychological health, based on evidence from meta-analyses,   is shown in Table 1.

The   search strategy was also detailed in the text [page 5, lines 102 – 105]:

“Meta-analyses were   identified using search terms “lifestyle” or “diet” or “exercise” and “PCOS”   using PubMed, Scopus and the Cochrane database of systematic reviews, through   expert referral and previous reviews with key publications selected for their   relevance by first authors.”

The title of Table 1   has been updated to:

A summary of current evidence from   meta-analyses on lifestyle management in PCOS”

Regarding   the role of healthcare professionals, you should note the following   reference:

A W   Lin, E J Bergomi, J S Dollahite, J Sobal, K M Hoeger, M E Lujan, Trust in   Physicians and Medical Experience Beliefs Differ Between Women With and   Without Polycystic Ovary Syndrome, Journal of the Endocrine Society, Volume 2, Issue 9, September 2018, Pages 1001–1009, https://doi.org/10.1210/js.2018-00181

This concludes women with PCOS reported   greater overall distrust in the PCP’s opinions and more arguments with their   health care providers than women without PCOS, which may influence some of   your recommendations

The healthcare professional section has been expanded   to include the dissatisfaction with healthcare professionals experienced by   women with PCOS [page 4, lines 186 – 191]:

“However, it should be noted women with PCOS have   reported dissatisfaction with provision of care and support from primary care   providers when compared to women without PCOS (32, 75). A 2018 study   concluded that women with PCOS report overall greater distrust in the   opinions of their primary care physicians and reportedly had more arguments with   other healthcare professionals, compared to women without PCOS (75). This may impact   the ability of PCOS care providers to effectively implement lifestyle   management.”

Reviewer 2 Report

The article seeks to review lifestyle management  in PCOS, specifically the barriers and facilitators to the  implementation of lifestyle management. It would be useful to have the  paper in double spaced format to make editing and review  easier. The authors review the aetiology of PCOS, the link with  obesity, and focus on the non-pharmacological management. The various  lifestyle interventions (diet, exercise, behavioural) are discussed, as  is the role of each different care provider. Table  1 includes a summary of the current evidence informing lifestyle  management in PCOS, and whilst there is significant heterogeneity in the  studies, it is a useful summary of the current evidence. The second  part of the paper discusses both the barriers and  facilitators to implementing lifestyle management.

The paper is relevant and timely given that PCOS  is a common condition, obesity is becoming an ever growing burden, and  lifestyle interventions have a proven benefit.

The paper is well written and thoroughly referenced, and I do not recommend any changes.

Author Response

Dear reviewer 2 and editor,

Thank you for the constructive feedback. Please see below our response to your comments and suggestions:

Comments and suggestions

Responses

Reviewer 2

The   article seeks to review lifestyle management  in PCOS, specifically the   barriers and facilitators to the  implementation of lifestyle   management. It would be useful to have the  paper in double spaced   format to make editing and review easier.

The text has been   reformatted with double-spacing.

The   authors review the aetiology of PCOS, the link with obesity, and focus on the   non-pharmacological management. The various lifestyle interventions (diet,   exercise, behavioural) are discussed, as is the role of each different   care provider. Table 1  includes a summary of the current evidence   informing lifestyle  management in PCOS, and whilst there is significant   heterogeneity in the  studies, it is a useful summary of the current   evidence.

Thank you.

The   second  part of the paper discusses both the barriers and    facilitators to implementing lifestyle management. The paper is   relevant and timely given that PCOS  is a common condition, obesity is   becoming an ever growing burden, and  lifestyle interventions have a   proven benefit.

The   paper is well written and thoroughly referenced, and I do not recommend any   changes.

Thank you.